# An Analysis of Ensemble Sampling

**Chao Qin**
Columbia University
cqin22@gsb.columbia.edu

**Zheng Wen  Xiuyuan Lu  Benjamin Van Roy**
DeepMind
{zhengwen,lxlu,benvanroy}@google.com

## Abstract

Ensemble sampling serves as a practical approximation to Thompson sampling when maintaining an exact posterior distribution over model parameters is computationally intractable. In this paper, we establish a regret bound that ensures desirable behavior when ensemble sampling is applied to the linear bandit problem. This represents the first rigorous regret analysis of ensemble sampling and is made possible by leveraging information-theoretic concepts and novel analytic techniques that may prove useful beyond the scope of this paper.

## 1 Introduction

Thompson sampling (TS) is a popular heuristic for balancing between exploration and exploitation in bandit learning. In its basic form, TS maintains a posterior distribution over model parameters. When the prior distribution and likelihood function exhibit conjugacy properties, the posterior distribution can be maintained through computationally tractable Bayesian inference. However, many practical contexts call for more complex models for which exact Bayesian inference becomes computationally intractable. For such contexts, ensemble sampling (ES) can serve as a practical approximation to TS [Lu and Van Roy, 2017].

Instead of the posterior distribution, ES maintains an ensemble of statistically plausible models that can be updated in an efficient incremental manner. The corresponding discrete distribution represents an approximation to the posterior distribution. At the start of each timestep, a model is sampled uniformly from the ensemble and an action is selected to optimize expected reward with respect to the sampled model. Each model is initially sampled from the prior distribution and evolves through an updating process that adapts to observations and random perturbations.

While a growing literature [Osband et al., 2016, Russo et al., 2018, Osband et al., 2018, Lu et al., 2018, Osband et al., 2019, Dwaracherla et al., 2020, Osband et al., 2022] presents applications of ES, there has been no rigorous theory that ensures desirable behavior similar to that enjoyed by TS. A regret bound for ES applied to linear-Gaussian bandits is provided by Lu and Van Roy [2017], but a flaw in the associated analysis has brought this result into question. Since the publication of that paper, several researchers have tried to remedy the analysis or otherwise establish performance guarantees for ES, but these efforts have gone without success.

We offer in this paper the first rigorous regret analysis of ES. Like Lu and Van Roy [2017], we study ES applied to linear-Gaussian bandits. This serves as a simple sanity check for the approach. We establish a Bayesian regret bound (Theorem 1) that consists of two terms. The first term is identical to a Bayesian regret bound established for TS. The second term accounts for posterior mismatch and can be made arbitrarily small by increasing the ensemble size. Our analysis leverages information-theoretic concepts and novel analytic techniques that may prove useful beyond the scope of this paper.

As a stepping stone in our analysis, we establish a general Bayesian regret bound that applies to *any* learning algorithm for linear bandits (Theorem 2). In particular, this regret bound is based on the Hellinger distance between the action-selection distribution specified by the learning algorithm and

36th Conference on Neural Information Processing Systems (NeurIPS 2022).

the posterior distribution of the optimal action at each timestep. We believe that this regret bound is of independent interest and might be useful in analyzing other learning algorithms, such as other approximate TS algorithms.

Note that for linear-Gaussian bandits considered in this paper, Kalman filtering offers a tractable means to exact inference and, as such, an approximation method like ES is not required. However, it is worth mentioning that the analysis in this paper also offers insight into more complex models that call for approximate inference. For example, consider a neural network with very wide hidden layers. A recent research thrust (e.g., Jacot et al. [2018], Lee et al. [2017]) highlights that, under suitable technical conditions, training such a neural network via stochastic gradient descent (SGD) approximates Gaussian process inference. Our result extends directly to finite-armed Gaussian process bandits. This suggests that ES applied with neural networks that are trained via SGD can be effective in complex bandit problems, though formalizing this extension requires significant technical work, which we leave for future research.

The remainder of this paper is organized as follows: Section 2 reviews related work, and Section 3 formulates the considered linear bandit problem and describes the ensemble sampling algorithm. Section 4 presents the main result of this paper: a Bayesian regret bound for ES applied to linear bandits. Before diving into the analysis, we define some notation used in this paper in Section 5. Then we motivate and derive a general regret bound in Section 6, and apply it to analyzing ES in Section 7. Finally, we conclude the paper and discuss future directions in Section 8.

## 2 Related work

Thompson sampling [Thompson, 1933, Chapelle and Li, 2011, Russo et al., 2018] is a commonly used heuristic for balancing exploration and exploitation in bandits [Lattimore and Szepesvári, 2020] and reinforcement learning [Sutton and Barto, 2018]. However, the basic form of Thompson sampling can be computationally intractable unless the prior distribution and likelihood function exhibit conjugacy properties. To overcome this computational challenge, variants of approximate versions of TS have been developed (see Chapter 5 of Russo et al. [2018]), including approximate TS using Laplace approximation [Chapelle and Li, 2011], bootstrapping [Eckles and Kapteln, 2019, Kveton et al., 2019], variational inference [Urteaga and Wiggins, 2018, Yu et al., 2020], Markov chain Monte Carlo [Casella and George, 1992, Roberts and Tweedie, 1996], hypermodels [Dwaracherla et al., 2020], and optimal transport [Zhang et al., 2019].

Ensemble sampling is an approximate version of Thompson sampling, formally proposed by Lu and Van Roy [2017]. It has been one of the most popular methods in approximate Bayesian inference and has wide applications in sequential decision making. For example, variations of ES have been applied to (deep) reinforcement learning [Osband et al., 2016, 2018, 2019], online recommendation [Lu et al., 2018, Hao et al., 2020, Zhu and Van Roy, 2021], multi-agent reinforcement learning [Dimakopoulou and Van Roy, 2018], behavioral sciences [Eckles and Kapteln, 2019], and marketing strategies [Yang et al., 2020]. Recently, Osband et al. [2022] has developed a testbed to compare and rank agents designed for uncertainty modeling, including variants of ensemble agents.

A lot of work has attempted to analyze ensemble sampling, but none of them has been successful. In their original paper, Lu and Van Roy [2017] try to analyze ES in linear-Gaussian bandits, but there is a mistake in the final step of the analysis where they compare the regret of ES to that of TS. In particular, in the proof of Lemma 8, the hypothetical actions selected by TS are defined on the histories of ES, instead of TS. Therefore, the cumulative regret incurred by these hypothetical TS actions is not equal to the cumulative regret of applying TS throughout all timesteps, while the paper incorrectly claims that they are equal.

In addition to Lu and Van Roy [2017], Phan et al. [2019] aims to analyze general approximate TS with approximation errors measured in $\alpha$-divergence for $K$-armed bandit. To demonstrate the main result, Phan et al. [2019] uses ES as an example, but also inherits the technical flaw in Lu and Van Roy [2017]. In addition, though Phan et al. [2019] shows that the approximate version of TS with uniform sampling achieves sublinear regret, this result mainly benefits from this forced exploration.

There are also papers aiming to analyze bandit algorithms with approximate inference. A recent paper [Huang et al., 2022] studies an upper-confidence-bound (UCB) type algorithm for bandit problems with approximate inference. Another recent paper [Ash et al., 2021] proposes and analyzes

an ensemble based UCB algorithm for bandit problems. Kveton et al. [2020] studies linear bandits in the frequentist setting and provides a general regret bound that is tailored to analyzing randomized algorithms with good concentration and anti-concentration properties, but it could not be directly applied to analyzing ES due to ES's discrete and incremental update nature.

## 3 Preliminaries

We begin by introducing a formulation of the linear bandit and an ensemble sampling algorithm suited for this environment.

### 3.1 Linear bandit

The linear bandit [Abbasi-Yadkori et al., 2011, Lattimore and Szepesvári, 2020] has received much attention in the bandit learning literature. It serves as a simple didactic environment, often used to sanity-check agent designs. In this spirit, we will analyze ES as applied to the linear bandit.

We consider the linear bandit with a Gaussian prior distribution and likelihood function. An instance is characterized by a tuple $\mathcal{E} = (\mathcal{A}, \mu_0, \Sigma_0, \sigma^2)$, where $\mathcal{A} \subset \mathbb{R}^d$ is a finite action set, $\mu_0$ and $\Sigma_0$ are the prior mean vector and covariance matrix, and $\sigma^2$ is the noise variance. Let $K = |\mathcal{A}|$ denote the cardinality of the action set. Note that actions are vectors of dimension $d$. The prior distribution over the *unknown* coefficient vector $\theta \in \mathbb{R}^d$, $\mathbb{P}(\theta \in \cdot)$, is multivariate Gaussian $N(\mu_0, \Sigma_0)$. Rewards are generated by a random sequence $(R_t : t = 1, 2, \ldots)$ of $K$-dimensional vectors, which is i.i.d. conditioned on $\theta$. In particular, for $t = 1, 2 \ldots,$

$$R_{t,a} = a^\top \theta + W_{t,a} \quad \forall a \in \mathcal{A},$$

where $W_t = (W_{t,a})_{a \in \mathcal{A}}$ is a $K$-dimensional vector with each element distributed as $N(0, \sigma^2)$.

An agent interacts with the linear bandit as follows: at timestep $t = 0, 1, \ldots$, the agent executes an action $A_t$; then it observes a reward $R_{t+1,A_t}$. Note that only the reward associated with the executed action $A_t$ is observed through what is termed *bandit feedback*. The agent's experience through timestep $t$ is encoded by a history $H_t = (A_0, R_{1,A_0}, \ldots, A_{t-1}, R_{t,A_{t-1}})$. The agent's objective is to maximize expected reward over some duration $T$:

$$\sum_{t=0}^{T-1} \mathbb{E}\big[R_{t+1,A_t}\big].$$

This is equivalent to minimizing the Bayesian regret:

$$\text{Regret}(T) = \sum_{t=0}^{T-1} \mathbb{E}[R_{t+1,A_*} - R_{t+1,A_t}]$$

where $A_* \sim \text{unif}\big\{\arg\max_{a \in \mathcal{A}} a^\top \theta\big\}$. The expectations in both equations integrate over random reward realizations, algorithmic randomness, and the coefficient vector $\theta$. Note that $A_*$ is random since it depends on random $\theta$ and the uniform sampling among maximizers (which breaks ties).

### 3.2 Ensemble sampling

In the absence of conjugacy properties that enable efficient Bayesian inference, TS in its exact form becomes computationally infeasible. ES serves as a practical approximation to TS [Lu and Van Roy, 2017], often suitable for such contexts. The key idea behind ES is to maintain an ensemble of statistically plausible models that can be updated in an efficient incremental manner and to treat the discrete distribution represented by this ensemble of models as an approximation to the posterior distribution. The algorithm begins by sampling $M$ models from the prior distribution, where $M$ is a hyperparameter. At the start of each timestep, a model is sampled uniformly from the ensemble and an action is selected to optimize the immediate expected reward with respect to the sampled model. Then, each model is updated incrementally based on the observed reward and a random perturbation. Algorithm 1 presents the associated pseudo-code for ES.

**Algorithm 1** Ensemble Sampling
___
1: **Input**: $M$ and $\mathbb{P}(\theta \in \cdot)$
2: **Sample**: $\tilde{\theta}_{0,1}, \ldots, \tilde{\theta}_{0,M} \sim \mathbb{P}(\theta \in \cdot)$
3: **for** $t = 0, 1, \ldots$ **do**
4:      **Sample**: $m_t \sim \text{unif}\{1, \ldots, M\}$
5:      **Execute**: $A_t \sim \text{unif}\left\{ \arg\max_{a \in \mathcal{A}} a^\top \tilde{\theta}_{t,m_t} \right\}$
6:      **Observe**: $R_{t+1,A_t}$
7:      **Update**: $\tilde{\theta}_{t,m} \longrightarrow \tilde{\theta}_{t+1,m} \quad \forall m \in [M]$
8: **end for**
___

For Gaussian prior $N(\mu_0, \Sigma_0)$, by conjugacy, the posterior distribution is also Gaussian with covariance matrix and mean vector updated as follows,

$$\Sigma_{t+1} = \left( \Sigma_t^{-1} + \frac{1}{\sigma^2} A_t A_t^\top \right)^{-1} \quad \text{and} \quad \mu_{t+1} = \Sigma_{t+1}\left( \Sigma_t^{-1} \mu_t + \frac{R_{t+1,A_t}}{\sigma^2} A_t \right). \quad (1)$$

While this highlights conjugacy properties of a sort that allow for efficient Bayesian inference and thus tractable implementation of exact TS, the linear bandit with a Gaussian prior distribution and likelihood function serves as a useful context for studying the behavior of ES in relation to TS. Like [Lu and Van Roy, 2017], we consider a version of ES that updates each $m$-th model according to

$$\tilde{\theta}_{t+1,m} = \Sigma_{t+1}\left( \Sigma_t^{-1} \tilde{\theta}_{t,m} + \frac{R_{t+1,A_t} + \tilde{W}_{t+1,m}}{\sigma^2} A_t \right) \quad (2)$$

where $\tilde{W}_{t+1} = (\tilde{W}_{t+1,m})_{m \in [M]}$ is an $M$-dimensional vector with each element distributed as $N(0, \sigma^2)$ and $[M] = \{1, \ldots, M\}$.

## 4 A regret bound for ensemble sampling

In this section, we establish the following regret bound for ES (Algorithm 1) when it is applied to linear bandits described in Section 3.1.

**Theorem 1.** *Under ensemble sampling,*

$$\text{Regret}(T) \leq \underbrace{\iota\sqrt{dT\mathbb{H}(A_*)}}_{(a)} + \underbrace{\eta T \sqrt{\frac{K\log(6TM)}{M}}}_{(b)}$$

*where $\mathbb{H}(A_*)$ is the Shannon entropy of the optimal action $A_*$ under the prior, and*

$$\iota \triangleq \sqrt{2\left( \max_{a \in \mathcal{A}} a^\top \Sigma_0 a + \sigma^2 \right)} \quad \text{and} \quad \eta \triangleq 2\sqrt{\mathbb{E}\left[ \max_{a \in \mathcal{A}} (a^\top \theta)^2 \right] + \sigma^2}. \quad (3)$$

Note that $\iota = O(1)$ and Lemma 7 in Appendix A shows $\eta = O(\sqrt{\min\{d, \log K\}})$, so

$$\text{Regret}(T) \leq O\left( \sqrt{dT\mathbb{H}(A_*)} + T\sqrt{\frac{\min\{d, \log K\}K\log(6TM)}{M}} \right).$$

Theorem 1 represents the first rigorously proved regret bound that ensures some degree of robustness in application of ES to a nontrivial bandit problem. One limitation is that the second term $(b)$ in our regret bound depends on the action set size $K$ since our current analysis uses the method of (finite) types [Cover and Thomas, 2006], which might not be tight for linear bandit. We conjecture that the second term $(b)$ can be improved to be only dependent of the dimension $d$. Another limitation is that the result applies only to the linear bandit with Gaussian prior distribution and likelihood function, which admits tractable application of exact TS. Nevertheless, this result represents a beginning of the theory of ES, suggesting that the approach is sound and the analysis might be extended to broader problem classes.

**Comparison with the regret bound for TS** The regret bound in Theorem 1 consists of two terms. The first term $(a)$ is exactly the regret bound achieved by TS with exact posterior [Russo and Van Roy, 2016]. Since the action set size is $K$, the entropy of the optimal action $\mathbb{H}(A_*) \leq \log K$, and as discussed in Russo and Van Roy [2016], when the prior is informative, $\mathbb{H}(A_*) \ll \log K$. Therefore, the first term $(a)$ is order optimal and improves upon the worst-case regret bound achieved by other algorithms, e.g., upper confidence bound type algorithms. On the other hand, the second term $(b)$ is an incremental term and accounts for posterior mismatch. Note that as the ensemble size $M$ approaches infinity, the second term $(b)$ converges to zero and our regret bound for ES reduces to that for TS with exact posterior. Moreover, as long as the ensemble size $M$ reaches $KT/d$, our regret bound for ES has the same order as that for TS in terms of $T$ and $d$ (up to logarithmic factors).

**Comparison with the regret bound in Lu and Van Roy [2017]** Theorem 1 presents a Bayesian regret bound for the prior $N(\mu_0, \Sigma_0)$ over $\theta$, while Lu and Van Roy [2017] studies ES in the frequentist setting where unknown $\theta$ is fixed. They try to upper bound the frequentist regret of ES by that of TS plus some cumulative approximation error due to posterior mismatch, similar to the second term $(b)$ in our Bayesian regret bound. Although there are some technical issues in their analysis, as is discussed in Section 2, we compare both regret bounds. The frequentist regret bound in Theorem 3 of Lu and Van Roy [2017] suggests that for any $\epsilon > 0$, when the ensemble size $M = \tilde{\Theta}(K/\epsilon^2)$ ($\tilde{\Theta}$ hides logarithmic factors), their cumulative approximation error is $\epsilon \Delta(\theta) T$ where $\Delta(\theta) = \max_{a \in \mathcal{A}} a^\top \theta - \min_{a \in \mathcal{A}} a^\top \theta$. To make the second term $(b)$ in our regret bound scale as $\tilde{O}(\epsilon T)$ ($\tilde{O}$ hides logarithmic factors), we need the ensemble size $M$ to be $\tilde{\Theta}(K/\epsilon^2)$ as well. We conjecture the required ensemble size can be improved to be only dependent of the dimension $d$.

## 5 Information measures

Before proceeding, we present several information measures including Kullback–Leibler divergence, Hellinger distance, Shannon entropy and mutual information.

### 5.1 Kullback–Leibler divergence and Hellinger distance

For two probability distributions $P$ and $Q$, the Kullback–Leibler (KL) divergence between $P$ and $Q$ is defined as

$$\mathbf{d}_{\mathrm{KL}}(P\|Q) \triangleq \int \log\left(\frac{\mathrm{d}P}{\mathrm{d}Q}\right) \mathrm{d}P$$

if $P$ is absolutely continuous with respect to $Q$; otherwise, define $\mathbf{d}_{\mathrm{KL}}(P\|Q) = +\infty$. Note that $\mathbf{d}_{\mathrm{KL}}(P\|Q) \neq \mathbf{d}_{\mathrm{KL}}(Q\|P)$ in general. The Hellinger distance between $P$ and $Q$ is defined as

$$\mathbf{d}_{\mathrm{H}}(P\|Q) \triangleq \sqrt{\int \left(\sqrt{\mathrm{d}P} - \sqrt{\mathrm{d}Q}\right)^2}$$

which is symmetric in $P$ and $Q$. In the special case of discrete distributions $P = (p_1, \ldots, p_n)$ and $Q = (q_1, \ldots, q_n)$, the KL divergence and Hellinger distance between $P$ and $Q$ can be written as

$$\mathbf{d}_{\mathrm{KL}}(P\|Q) = \sum_{i \in [n]} p_i \log\left(p_i/q_i\right) \quad \text{and} \quad \mathbf{d}_{\mathrm{H}}(P\|Q) = \sqrt{\sum_{i \in [n]} \left(\sqrt{p_i} - \sqrt{q_i}\right)^2}.$$

### 5.2 Shannon entropy and mutual information

Consider a random variable $X$ that takes values in a countable set $\mathcal{X}$. The *Shannon entropy* of $X$ is

$$\mathbb{H}(X) \triangleq -\sum_{x \in \mathcal{X}} \mathbb{P}(X = x) \log \mathbb{P}(X = x) = \mathbb{E}_X[-\log \mathbb{P}(X \in \cdot)]$$

with a convention that $0 \log 0 = 0$.

For ease of exposition, let $Y$ be another random variable that takes values in a countable set $\mathcal{Y}$. For $y \in \mathcal{Y}$ such that $\mathbb{P}(Y = y) > 0$, the *realized conditional entropy* of $X$ given $Y = y$ is

$$\mathbb{H}(X|Y = y) \triangleq -\sum_{x \in \mathcal{X}} \mathbb{P}(X = x|Y = y) \log \mathbb{P}(X = x|Y = y) = \mathbb{E}_X[-\log \mathbb{P}(X \in \cdot|Y)|Y = y],$$

and the *conditional entropy* of $X$ given $Y$ is
$$\mathbb{H}(X|Y) \triangleq \mathbb{E}_Y\left[-\sum_{x \in \mathcal{X}} \mathbb{P}(X = x|Y)\log \mathbb{P}(X = x|Y)\right] = \mathbb{E}_Y[\mathbb{H}(X|Y = Y)].$$
Note that the above definitions can be extended to general random variables.

The *mutual information* between $X$ and $Y$ is
$$\mathbb{I}(X;Y) \triangleq \mathbf{d}_{\mathrm{KL}}\left(\mathbb{P}(X \in \cdot, Y \in \cdot) \,\|\, \mathbb{P}(X \in \cdot)\mathbb{P}(Y \in \cdot)\right) = \mathbb{H}(X) - \mathbb{H}(X|Y),$$
and the *conditional mutual information* between $X$ and $Y$ given a general random variable $Z$ is
$$\mathbb{I}(X;Y|Z) \triangleq \mathbb{E}_Z\left[\mathbf{d}_{\mathrm{KL}}\left(\mathbb{P}(X \in \cdot, Y \in \cdot|Z) \,\|\, \mathbb{P}(X \in \cdot|Z)\mathbb{P}(Y \in \cdot|Z)\right)\right] = \mathbb{H}(X|Z) - \mathbb{H}(X|Y, Z).$$

# 6 A general regret bound

In this section, we derive a general regret bound for *any* learning algorithm (Theorem 2), based on the Hellinger distance between the (conditional) action-selection distribution specified by the algorithm and the posterior distribution of the optimal action $A_*$, and apply it to analyzing the regret bound for ES in Theorem 1. We believe that our regret bound is of independent interest and might be used to analyze other bandit algorithms, such as variants of approximate TS algorithms, in the future.

To simplify the exposition, we first define some shorthand notation to simplify the exposition of this paper. First, we use subscript $t$ to denote conditioning on $H_t$, specifically, we define
$$\mathbb{P}_t(\cdot) \triangleq \mathbb{P}(\cdot|H_t) \quad \text{and} \quad \mathbb{E}_t[\cdot] \triangleq \mathbb{E}[\cdot|H_t].$$
In addition, we define two probability distributions $\overline{p}_t$ and $p_t$ over action set $\mathcal{A}$ as
$$\overline{p}_t(\cdot) \triangleq \mathbb{P}_t(A_t = \cdot) \quad \text{and} \quad p_t(\cdot) \triangleq \mathbb{P}_t(A_* = \cdot).$$
Conditional on the history $H_t$, $\overline{p}_t$ is the sampling distribution of the action $A_t$ and $p_t$ is the posterior distribution of the optimal action $A_*$. Both $p_t$ and $\overline{p}_t$ are specific to the algorithm. Note that given the history $H_t$, $\overline{p}_t$ and $p_t$ are the same under TS, but they are not under other algorithms. For approximate TS algorithms, if the approximation is accurate, $p_t$ is close to $\overline{p}_t$. Now we introduce a general regret bound for any learning algorithm:

**Theorem 2.** *Under any learning algorithm,*
$$\mathrm{Regret}(T) \leq \iota\sqrt{dT\mathbb{H}(A_*)} + \eta \sum_{t=0}^{T-1} \sqrt{\mathbb{E}\left[\mathbf{d}_{\mathrm{H}}^2(\overline{p}_t\|p_t)\right]}$$
*where $\iota$ and $\eta$ are defined in Equation (3).*

The regret bound in Theorem 2 holds for any learning algorithm. Note $\mathbb{H}(A_*)$ is the entropy of the optimal action $A_*$ and we always have $\mathbb{H}(A_*) \leq \log K$. The first term matches the regret bound for TS derived in Russo and Van Roy [2016]. In the second term, $\sum_{t=0}^{T-1} \sqrt{\mathbb{E}\left[\mathbf{d}_{\mathrm{H}}^2(\overline{p}_t\|p_t)\right]}$ quantifies the cumulative distance between the sampling distribution of the action $A_t$ and the posterior distribution the optimal action $A_*$. Under TS with exact posterior, $\mathbf{d}_{\mathrm{H}}^2(\overline{p}_t\|p_t) = 0$ due to the posterior matching property of TS, i.e., $\overline{p}_t = p_t$. Moreover, for approximate TS algorithms, if the approximation is accurate, $\mathbf{d}_{\mathrm{H}}^2(\overline{p}_t\|p_t)$ should be small.

**Regret bounds in terms of KL divergence between $\overline{p}_t$ and $p_t$** It is sometimes more convenient to analyze the KL divergence between $\overline{p}_t$ and $p_t$. Recall that the squared Hellinger distance is symmetric and upper bound by the KL divergence.

**Fact 1** (Lemma 2.4 in Tsybakov [2009]). *For two probability distributions $P$ and $Q$, we have*
$$\mathbf{d}_{\mathrm{H}}^2(P\|Q) \leq \min\{\mathbf{d}_{\mathrm{KL}}(P\|Q), \mathbf{d}_{\mathrm{KL}}(Q\|P)\}.$$

We obtain the regret upper bound with the approximation error in terms of two forms of KL divergence between $\overline{p}_t$ and $p_t$.

**Corollary 1.** *Under any learning algorithm,*
$$\mathrm{Regret}(T) \leq \iota\sqrt{dT\mathbb{H}(A_*)} + \eta \sum_{t=0}^{T-1} \sqrt{\mathbb{E}\left[\min\{\mathbf{d}_{\mathrm{KL}}(\overline{p}_t\|p_t), \mathbf{d}_{\mathrm{KL}}(p_t\|\overline{p}_t)\}\right]}.$$

We will show in Section 7 that for ensemble sampling, $\mathbf{d}_{\mathrm{KL}}(\overline{p}_t\|p_t)$ can be bounded in terms of the ensemble size $M$. It is noteworthy that in some other problems, it is more convenient to bound the other form $\mathbf{d}_{\mathrm{KL}}(p_t\|\overline{p}_t)$. One such example is the regret bound developed in Wen et al. [2022].

## 6.1 Proof of Theorem 2

For $t = 0, 1, \ldots,$ we define *per-timestep main regret* $G_t$ and *per-timestep approximation error* $D_t$:

$$G_t \triangleq \sum_{a \in \mathcal{A}} \sqrt{\overline{p}_t(a) p_t(a)} \left( \mathbb{E}_t[R_{t+1,a}|A_* = a] - \mathbb{E}_t[R_{t+1,a}] \right),$$

$$D_t \triangleq \sum_{a \in \mathcal{A}} \left( \sqrt{p_t(a)} - \sqrt{\overline{p}_t(a)} \right) \left( \sqrt{p_t(a)} \mathbb{E}_t[R_{t+1,a}|A_* = a] + \sqrt{\overline{p}_t(a)} \mathbb{E}_t[R_{t+1,a}] \right).$$

**Lemma 1.** *The cumulative regret can be written as follows,*

$$\text{Regret}(T) = \sum_{t=0}^{T-1} \mathbb{E}[G_t + D_t].$$

*Proof.* By the tower property of conditional expectation,

$$\text{Regret}(T) = \sum_{t=0}^{T-1} \mathbb{E}\left[ \mathbb{E}_t[R_{t+1,A_*} - R_{t+1,A_t}] \right]$$

where

$$\mathbb{E}_t[R_{t+1,A_*} - R_{t+1,A_t}] = \sum_{a \in \mathcal{A}} p_t(a) \mathbb{E}_t[R_{t+1,a}|A_* = a] - \sum_{a \in \mathcal{A}} \overline{p}_t(a) \mathbb{E}_t[R_{t+1,a}|A_t = a]$$

$$= \sum_{a \in \mathcal{A}} p_t(a) \mathbb{E}_t[R_{t+1,a}|A_* = a] - \sum_{a \in \mathcal{A}} \overline{p}_t(a) \mathbb{E}_t[R_{t+1,a}] = G_t + D_t$$

where the second equality uses that $R_{t+1} = (R_{t+1,a})_{a \in \mathcal{A}}$ is independent of the action $A_t$. □

Next we upper bound $\sum_{t=0}^{T-1} \mathbb{E}[G_t]$ and $\sum_{t=0}^{T-1} \mathbb{E}[D_t]$, respectively. We first analyze per-timestep main regret $G_t$. The following lemma generalizes Proposition 5 and Corollary 1 in Russo and Van Roy [2016] for the case such that $A_t$ and $A_*$ are not identically distributed given the history.

**Lemma 2.** *With $\iota$ defined in Equation (3), for $t = 0, 1 \ldots, T - 1$,*

$$G_t \leq \iota \sqrt{d \cdot \mathbb{I}_t(A_*; (A_t, R_{t+1,A_t}))}$$

*where $\mathbb{I}_t(A_*; (A_t, R_{t+1,A_t}))$ is the mutual information between the optimal action $A_*$ and action-reward pair $(A_t, R_{t+1,A_t})$ conditioning on a given history $H_t$ at timestep $t$.*

*Proof.* Let the action set $\mathcal{A} = \{a_1, \ldots, a_K\}$. Fix $t$ and define $M \in \mathbb{R}^{K \times K}$ by

$$M_{i,j} = \sqrt{\overline{p}_t(a_i) p_t(a_j)} \left( \mathbb{E}_t[R_{t+1,a_i}|A_* = a_j] - \mathbb{E}_t[R_{t+1,a_i}] \right) \quad \forall i, j \in [K].$$

Then $G_t = \text{Trace}(M)$. By the proof of Proposition 2 in Russo and Van Roy [2016],

$$\mathbb{I}_t(A_*; (A_t, R_{t+1,A_t})) = \mathbb{I}_t(A_*; A_t) + \mathbb{I}_t(A_*; R_{t+1,A_t}|A_t)$$

$$= \mathbb{I}_t(A_*; R_{t+1,A_t}|A_t)$$

$$= \sum_{i \in [K]} \overline{p}_t(a_i) \mathbb{I}_t(A_*; R_{t+1,A_t}|A_t = a_i)$$

$$= \sum_{i \in [K]} \overline{p}_t(a_i) \mathbb{I}_t(A_*; R_{t+1,a_i})$$

$$= \sum_{i \in [K]} \overline{p}_t(a_i) \left( \sum_{j \in [K]} p_t(a_j) \mathbf{d}_{\text{KL}} \left( \mathbb{P}_t(R_{t+1,a_i} \in \cdot | A_* = a_j) \| \mathbb{P}_t(R_{t+1,a_i} \in \cdot) \right) \right)$$

$$= \sum_{i,j \in [K]} \overline{p}_t(a_i) p_t(a_j) \mathbf{d}_{\text{KL}} \left( \mathbb{P}_t(R_{t+1,a_i} \in \cdot | A_* = a_j) \| \mathbb{P}_t(R_{t+1,a_i} \in \cdot) \right)$$

where the first equality uses the chain rule of mutual information (Fact 2); the second and fourth ones follow from that conditional on the history $H_t$, $A_t$ is jointly independent of $A_*$ and $R_{t+1} = (R_{t+1,a})_{a \in \mathcal{A}}$; the fifth equality uses the KL divergence form of mutual information (Fact 3).

By the updating rule on covariance matrix in Equation (1), conditional on $H_t$, $\theta \sim N(\mu_t, \Sigma_t)$ with $\Sigma_t \preceq \Sigma_0$. Hence, for any $i \in [K]$,

$$R_{t+1,a_i} = a_i^\top \theta + W_{t,a_i} \sim N\left(a_i^\top \mu_t, a_i^\top \Sigma_t a_i + \sigma^2\right) \quad \text{where} \quad a_i^\top \Sigma_t a_i \leq a_i^\top \Sigma_0 a_i \leq \max_{a \in \mathcal{A}} a^\top \Sigma_0 a.$$

This implies that $R_{t+1,a_i}$ is sub-Gaussian with variance proxy $(\max_{a \in \mathcal{A}} a^\top \Sigma_0 a + \sigma^2)$. Then by Lemma 10, we have

$$
\begin{aligned}
\mathbb{I}_t(A_*; (A_t, R_{t+1,A_t})) &\geq \frac{\sum_{i,j \in [K]} \overline{p}_t(a_i) p_t(a_j) \left(\mathbb{E}_t[R_{t+1,a_i}|A_* = a_j] - \mathbb{E}_t[R_{t+1,a_i}]\right)^2}{2\left(\max_{a \in \mathcal{A}} a^\top \Sigma_0 a + \sigma^2\right)} \\
&= \frac{\sum_{i,j \in [K]} M_{i,j}^2}{2\left(\max_{a \in \mathcal{A}} a^\top \Sigma_0 a + \sigma^2\right)} \\
&= \frac{\|M\|_{\mathrm{F}}^2}{2\left(\max_{a \in \mathcal{A}} a^\top \Sigma_0 a + \sigma^2\right)}.
\end{aligned}
$$

Then by Fact 4,

$$\frac{G_t^2}{\mathbb{I}_t(A_*; (A_t, R_{t+1,A_t}))} \leq \frac{2\left(\max_{a \in \mathcal{A}} a^\top \Sigma_0 a + \sigma^2\right) \mathrm{Trace}(M)^2}{\|M\|_{\mathrm{F}}^2} \leq 2\left(\max_{a \in \mathcal{A}} a^\top \Sigma_0 a + \sigma^2\right) \mathrm{Rank}(M).$$

Now we show that $\mathrm{Rank}(M) \leq d$. Define

$$\nu = \mathbb{E}_t[\theta] \quad \text{and} \quad \nu_j = \mathbb{E}_t[\theta|A_* = a_j] \quad \forall j \in [K].$$

Then for any $i, j \in [K]$,

$$M_{i,j} = \sqrt{\overline{p}_t(a_i) p_t(a_j)} \left(\mathbb{E}_t[\theta^\top a_i | A_* = a_j] - \mathbb{E}_t[\theta^\top a_i]\right) = \sqrt{\overline{p}_t(a_i) p_t(a_j)} (\nu_j - \nu)^\top a_i,$$

and thus

$$M = \begin{bmatrix} \sqrt{p_t(a_1)} (\nu_1 - \nu)^\top \\ \vdots \\ \sqrt{p_t(a_K)} (\nu_K - \nu)^\top \end{bmatrix} \begin{bmatrix} \sqrt{\overline{p}_t(a_1)} a_1 & \cdots & \sqrt{\overline{p}_t(a_K)} a_K \end{bmatrix}.$$

Since $M$ is the product of one $K \times d$ matrix and the other $d \times K$ matrix, we have $\mathrm{Rank}(M) \leq d$. □

Now we are ready to bound $\sum_{t=0}^{T-1} \mathbb{E}[G_t]$ and derive the first term of the regret bound in Theorem 2 by following the analysis of Proposition 1 in Russo and Van Roy [2016].

**Lemma 3.** *With $\iota$ defined in Equation (3), the following inequality holds:*

$$\sum_{t=0}^{T-1} \mathbb{E}[G_t] \leq \iota \sqrt{dT\mathbb{H}(A_*)}.$$

*Proof.* By Lemma 2,

$$
\begin{aligned}
\sum_{t=0}^{T-1} \mathbb{E}[G_t] \leq \iota \sqrt{d} \sum_{t=0}^{T-1} \mathbb{E}\left[\sqrt{\mathbb{I}_t(A_*, (A_t; R_{t+1,A_t}))}\right] &\leq \iota \sqrt{dT\mathbb{E}\left[\sum_{t=0}^{T-1} \mathbb{I}_t(A_*; (A_t, R_{t+1,A_t}))\right]} \\
&= \iota \sqrt{dT \sum_{t=0}^{T-1} \mathbb{I}(A_*; (A_t, R_{t+1,A_t})|H_t)} \\
&= \iota \sqrt{dT \left(\mathbb{H}(A_*) - \mathbb{H}(A_*|H_T)\right)} \\
&\leq \iota \sqrt{dT\mathbb{H}(A_*)}
\end{aligned}
$$

where the second inequality applies the Cauchy-Schwarz inequality, and the last inequality uses the chain rule for mutual information (Fact 2) and the definition of mutual information. □

The next lemma controls the expected per-timestep approximation error $\mathbb{E}[D_t]$ under (sub-)Gaussian prior distribution and noises. Summing over timesteps completes the proof of Theorem 2.

**Lemma 4.** *With $\eta$ defined in Equation (3), for $t = 0, 1, \ldots, T - 1$,*

$$\mathbb{E}[D_t] \leq \eta \sqrt{\mathbb{E}\left[\mathbf{d}_{\mathrm{H}}^2(\overline{p}_t \| p_t)\right]}.$$

*Proof.* Fix $t$. We write

$$D_t = \sum_{a \in \mathcal{A}} \left(\sqrt{p_t(a)} - \sqrt{\overline{p}_t(a)}\right) \sqrt{p_t(a)} \mathbb{E}_t[R_{t+1,a} | A_* = a] + \sum_{a \in \mathcal{A}} \left(\sqrt{p_t(a)} - \sqrt{\overline{p}_t(a)}\right) \sqrt{\overline{p}_t(a)} \mathbb{E}_t[R_{t+1,a}]$$

$$\leq \sqrt{\sum_{a \in \mathcal{A}} \left(\sqrt{p_t(a)} - \sqrt{\overline{p}_t(a)}\right)^2} \left(\sqrt{\sum_{a \in \mathcal{A}} p_t(a) \left(\mathbb{E}_t[R_{t+1,a} | A_* = a]\right)^2} + \sqrt{\sum_{a \in \mathcal{A}} \overline{p}_t(a) \left(\mathbb{E}_t[R_{t+1,a}]\right)^2}\right)$$

$$\leq \mathbf{d}_{\mathrm{H}}(\overline{p}_t \| p_t) \left(\sqrt{\sum_{a \in \mathcal{A}} p_t(a) \left(\mathbb{E}_t\left[R_{t+1,a}^2 | A_* = a\right]\right)} + \sqrt{\sum_{a \in \mathcal{A}} \overline{p}_t(a) \left(\mathbb{E}_t\left[R_{t+1,a}^2\right]\right)}\right)$$

$$= \mathbf{d}_{\mathrm{H}}(\overline{p}_t \| p_t) \left(\sqrt{\sum_{a \in \mathcal{A}} p_t(a) \mathbb{E}_t\left[R_{t+1,a}^2 | A_* = a\right]} + \sqrt{\sum_{a \in \mathcal{A}} \overline{p}_t(a) \mathbb{E}_t\left[R_{t+1,a}^2 | A_t = a\right]}\right)$$

$$= \mathbf{d}_{\mathrm{H}}(\overline{p}_t \| p_t) \left(\sqrt{\mathbb{E}_t\left[R_{t+1,A_*}^2\right]} + \sqrt{\mathbb{E}_t\left[R_{t+1,A_t}^2\right]}\right)$$

where the first inequality applies the Cauchy–Schwarz inequality; the second inequality uses the definition of Hellinger distance and the Jensen's inequality; the second-to-last equality holds since $A_t$ is independent of $R_{t+1} = (R_{t+1,a})_{a \in \mathcal{A}}$. Taking expectation of both sides, we have

$$\mathbb{E}[D_t] \leq \mathbb{E}\left[\mathbf{d}_{\mathrm{H}}(\overline{p}_t \| p_t) \sqrt{\mathbb{E}_t\left[R_{t+1,A_*}^2\right]}\right] + \mathbb{E}\left[\mathbf{d}_{\mathrm{H}}(\overline{p}_t \| p_t) \sqrt{\mathbb{E}_t\left[R_{t+1,A_t}^2\right]}\right]$$

$$\leq \sqrt{\mathbb{E}\left[\mathbf{d}_{\mathrm{H}}^2(\overline{p}_t \| p_t)\right]} \left(\sqrt{\mathbb{E}\left[R_{t+1,A_*}^2\right]} + \sqrt{\mathbb{E}\left[R_{t+1,A_t}^2\right]}\right)$$

$$= \sqrt{\mathbb{E}\left[\mathbf{d}_{\mathrm{H}}^2(\overline{p}_t \| p_t)\right]} \left(\sqrt{\mathbb{E}\left[(A_*^\top \theta + W_{t+1,A_*})^2\right]} + \sqrt{\mathbb{E}\left[(A_t^\top \theta + W_{t+1,A_t})^2\right]}\right)$$

$$\leq \sqrt{\mathbb{E}\left[\mathbf{d}_{\mathrm{H}}^2(\overline{p}_t \| p_t)\right]} \left(\sqrt{\mathbb{E}\left[(A_*^\top \theta)^2 + \sigma^2\right]} + \sqrt{\mathbb{E}\left[(A_t^\top \theta)^2\right] + \sigma^2}\right)$$

$$\leq 2 \sqrt{\mathbb{E}\left[\mathbf{d}_{\mathrm{H}}^2(\overline{p}_t \| p_t)\right]} \sqrt{\mathbb{E}\left[\max_{a \in \mathcal{A}} (a^\top \theta)^2\right] + \sigma^2}$$

where the second inequality uses the Cauchy–Schwarz inequality and tower property of conditional expectation, and the second-to-last inequality holds for the independent mean-zero (sub-)Gaussian noises with variance bounded by $\sigma^2$. $\square$

## 7 Completion of the proof of Theorem 1

In this section, we complete the proof of Theorem 1 by controlling the cumulative approximation error $\sum_{t=0}^{T-1} \sqrt{\mathbb{E}\left[\mathbf{d}_{\mathrm{KL}}(\overline{p}_t \| p_t)\right]}$ in Corollary 1 for ensemble sampling, where $\overline{p}_t$ is the sampling distribution of the action $A_t$ and $p_t$ is the posterior distribution of the optimal action $A_*$. It suffices to show Lemma 5 below for the per-timestep approximation error $\mathbb{E}[\mathbf{d}_{\mathrm{KL}}(\overline{p}_t \| p_t)]$ since summing over timesteps yields the second term of the regret bound for ensemble sampling in Theorem 1.

**Lemma 5.** *Under ensemble sampling, for $t = 0, 1, \ldots, T - 1$,*

$$\mathbb{E}\left[\mathbf{d}_{\mathrm{KL}}(\overline{p}_t \| p_t)\right] \leq \frac{K \log(6(t+1)M)}{M}.$$

*Proof.* We first discuss the behavior of ensemble sampling. Ensemble sampling uniformly samples a model $m \in \{1, \ldots, M\}$, and then chooses the action $A_t$ uniformly from the set $\tilde{\mathcal{A}}_{t,m} \triangleq$

$\arg\max_{a \in \mathcal{A}} a^\top \tilde{\theta}_{t,m}$. We define the following approximation of $p_t(a)$:
$$\hat{p}_t(a) \triangleq \frac{1}{M} \sum_{m \in [M]} \frac{1}{|\tilde{\mathcal{A}}_{t,m}|} \mathbb{1}\left\{ a \in \tilde{\mathcal{A}}_{t,m} \right\}$$
where $|\tilde{\mathcal{A}}_{t,m}|$ is the cardinality of the set $\tilde{\mathcal{A}}_{t,m}$ (with probability one, $|\tilde{\mathcal{A}}_{t,m}| = 1$). At timestep $t$, ES would sample an action from the approximation $\hat{p}_t$. Notice that the history $H_t$ does not include independent random perturbation $\tilde{W}_t = (\tilde{W}_{t,m})_{m \in [M]} \sim N(0, \sigma^2 I)$, which is used in updating $M$ models in Equation (2). Therefore, given history $H_t$, $\hat{p}_t(a)$ is still a random variable, and its expectation is the conditional probability of sampling action $a$, i.e., $\mathbb{E}_t[\hat{p}_t(a)] = \overline{p}_t(a)$. By convexity of KL divergence, the per-timestep approximation error
$$\mathbf{d}_{\mathrm{KL}}(\overline{p}_t \| p_t) = \mathbf{d}_{\mathrm{KL}}\left(\mathbb{E}_t[\hat{p}_t] \| p_t\right) \leq \mathbb{E}_t\left[\mathbf{d}_{\mathrm{KL}}(\hat{p}_t \| p_t)\right].$$
Taking expectation of both sides over $H_t$ gives $\mathbb{E}\left[\mathbf{d}_{\mathrm{KL}}(\overline{p}_t \| p_t)\right] \leq \mathbb{E}\left[\mathbf{d}_{\mathrm{KL}}(\hat{p}_t \| p_t)\right]$. Next we upper bound $\mathbb{E}\left[\mathbf{d}_{\mathrm{KL}}(\hat{p}_t \| p_t)\right]$ by first writing it in terms of the cumulative distribution function:
$$\mathbb{E}\left[\mathbf{d}_{\mathrm{KL}}(\hat{p}_t \| p_t)\right] = \int_0^\infty \mathbb{P}\left(\mathbf{d}_{\mathrm{KL}}(\hat{p}_t \| p_t) > \epsilon\right) \mathrm{d}\epsilon. \tag{4}$$
Then we use the following result to upper bound $\mathbb{P}\left(\mathbf{d}_{\mathrm{KL}}(\hat{p}_t \| p_t) > \epsilon\right)$ in the above integral.

**Lemma 6** (Lemma 4 in Lu and Van Roy [2017]). *Let $\epsilon > 0$. For $t = 0, 1, \ldots, T - 1$,*
$$\mathbb{P}\left(\mathbf{d}_{\mathrm{KL}}(\hat{p}_t \| p_t) > \epsilon \mid \theta\right) \leq (t+1)^K (M+1)^K e^{-M\epsilon}.$$

Taking expectation of both sides over $\theta$ gives the same upper bound on $\mathbb{P}\left(\mathbf{d}_{\mathrm{KL}}(\hat{p}_t \| p_t) > \epsilon\right)$, and thus for any $\delta \geq 0$, the integral in Equation (4) can be decomposed and bounded as follows,
$$\int_0^\infty \mathbb{P}\left(\mathbf{d}_{\mathrm{KL}}(\hat{p}_t \| p_t) > \epsilon\right) \mathrm{d}\epsilon = \int_0^\delta \mathbb{P}\left(\mathbf{d}_{\mathrm{KL}}(\hat{p}_t \| p_t) > \epsilon\right) \mathrm{d}\epsilon + \int_\delta^\infty \mathbb{P}\left(\mathbf{d}_{\mathrm{KL}}(\hat{p}_t \| p_t) > \epsilon\right) \mathrm{d}\epsilon$$
$$\leq \delta + (t+1)^K (M+1)^K \int_\delta^\infty e^{-M\epsilon} \mathrm{d}\epsilon$$
$$= \delta + \frac{(t+1)^K (M+1)^K e^{-M\delta}}{M}.$$
Taking derivative of RHS gives optimal $\delta^* = \frac{K[\log(t+1) + \log(M+1)]}{M}$, and then plugging in $\delta^*$ yields
$$\int_0^\infty \mathbb{P}\left(\mathbf{d}_{\mathrm{KL}}(\hat{p}_t \| p_t) > \epsilon\right) \mathrm{d}\epsilon \leq \frac{K[\log(t+1) + \log(M+1)] + 1}{M} \leq \frac{K \log(6(t+1)M)}{M}.$$
This completes the proof of Lemma 5, and summing over timesteps gives Theorem 1. $\qquad \square$

## 8 Concluding remarks

We present the first rigorous analysis for ensemble sampling, an approximate Thompson sampling method. ES maintains an ensemble of statistically plausible models that can be updated in an efficient incremental manner. We develop the regret bound for ES in Theorem 1 for linear bandits, which approaches the regret bound for TS as the ensemble size increases. The linear bandit problem serves only as a sanity check, and as discussed in Section 2, ES has been developed for broader and more challenging bandit and reinforcement learning problems. It is noteworthy that in many practical problems, ES with a moderate ensemble size (e.g. $\leq 30$ models) outperforms other state-of-the-art benchmarks [Lu and Van Roy, 2017, Lu et al., 2018, Osband et al., 2019].

As a stepping stone, we also establish a general regret bound in Theorem 2 that applies to *any* learning algorithm for linear bandits. This general regret bound may be of independent interest, since it is particularly useful for analyzing other varieties of approximate TS. One only need to bound the (cumulative) approximation error measured by the Hellinger distance between the action-selection distribution specified by the algorithm and the posterior distribution of the optimal action.

Finally, Theorem 2 is established by bounding the information ratios for linear bandits, and we believe it could be generalized to other bandit and online learning problems as long as the corresponding information ratios can be appropriately bounded. These problems include generalized linear bandits, combinatorial semi-bandits, online learning with full information feedback, and problems with pure exploration objectives. We also conjecture that the results of this paper can be extended to episodic reinforcement learning, but leave these extensions to future work.

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
