# Appendices

## A An upper bound $O(\sqrt{\min\{d, \log K\}})$ on $\eta$ in Theorem 1

In Equation (3), we define

$$\eta = 2\sqrt{\mathbb{E}\left[\max_{a \in \mathcal{A}}\left(a^\top \theta\right)^2\right] + \sigma^2}$$

where the expectation is taken with respect to the prior distribution over the coefficient vector $\theta$. In the following result, we derive an upper bound on $\eta$ for $\theta \sim N(\mu_0, \Sigma_0)$ by using that the expectation of the maximum of $K$ squares of (potentially correlated) (sub-)Gaussian random variables is $O(\log K)$. Note that this upper bound also holds for any sub-Gaussian prior distribution.

**Lemma 7.** *The following upper bound on $\eta$ holds:*

$$\eta \leq 2\sqrt{\min\left\{d \cdot \max_{a \in \mathcal{A}} \|a\|^2 \max_{i \in [d]}\left(\mu_{0,i}^2 + \Sigma_{0,i,i}\right), (4\log K + 5)\max_{a \in \mathcal{A}} a^\top \Sigma_0 a + \max_{a \in \mathcal{A}}\left(a^\top \mu_0\right)^2\right\} + \sigma^2}$$

$$= O(\sqrt{\min\{d, \log K\}}),$$

*where for any $i \in [d]$, $\mu_{0,i}$ is the $i$-th element of $\mu_0$ and $\Sigma_{0,i,i}$ is the $i$-th diagonal element of $\Sigma_0$.*

*Proof.* Using the Cauchy-Schwarz inequality, we have

$$\mathbb{E}\left[\max_{a \in \mathcal{A}}\left(a^\top \theta\right)^2\right] \leq \max_{a \in \mathcal{A}} \|a\|^2 \mathbb{E}\left[\|\theta\|^2\right] = \max_{a \in \mathcal{A}} \|a\|^2 \sum_{i \in [d]} \mathbb{E}\left[\theta_i^2\right] \leq d \cdot \max_{a \in \mathcal{A}} \|a\|^2 \max_{i \in [d]}\left(\mu_{0,i}^2 + \Sigma_{0,i,i}\right)$$

where $\theta_i$ is the $i$-th element in $\theta$, which follows $N(\mu_{0,i}, \Sigma_{0,i,i})$.

On the other hand, since $a^\top \theta \sim N\left(a^\top \mu_0, a^\top \Sigma_0 a + \sigma^2\right)$ for any $a \in \mathcal{A}$, applying Lemma 8 below gives

$$\mathbb{E}\left[\max_{a \in \mathcal{A}}\left(a^\top \theta\right)^2\right] \leq (4\log K + 5)\max_{a \in \mathcal{A}} a^\top \Sigma_0 a + \max_{a \in \mathcal{A}}\left(a^\top \mu_0\right)^2.$$

This completes the proof. $\square$

### A.1 Expectation of maximum of squares of sub-Gaussian random variables

We upper bound the expectation of maximum of squares of sub-Gaussian random variables.

**Lemma 8.** *For any $i \in [K]$, let $X_i$ be a sub-Gaussian random variable with mean $\nu_i$ and variance proxy $\sigma_i^2$. The following inequality holds:*

$$\mathbb{E}\left[\max_{i \in [K]} X_i^2\right] \leq (4\log K + 5)\max_{i \in [K]} \sigma_i^2 + \max_{i \in [K]} \nu_i^2.$$

*Proof.* For $\lambda > 0$,

$$\exp\left(\lambda \mathbb{E}\left[\max_{i \in [K]} X_i^2\right]\right) \leq \mathbb{E}\left[\exp\left(\lambda \max_{i \in [K]} X_i^2\right)\right] = \mathbb{E}\left[\max_{i \in [K]} \exp(\lambda X_i^2)\right] \leq \sum_{i \in [K]} \mathbb{E}[\exp(\lambda X_i^2)]$$

where the first inequality uses the Jensen's inequality. For $\lambda \in \left(0, \frac{1}{4\max_{i \in [K]} \sigma_i^2}\right]$, by applying Lemma 9 (Appendix B in Honorio and Jaakkola [2014]) below to the RHS above, we have

$$\exp\left(\lambda \mathbb{E}\left[\max_{i \in [K]} X_i^2\right]\right) \leq \sum_{i \in [K]} \exp(16\lambda^2 \sigma_i^4 + \lambda \mathbb{E}[X_i^2]) \leq K \exp\left(16\lambda^2 \max_{i \in [K]} \sigma_i^4 + \lambda \max_{i \in [K]} \mathbb{E}[X_i^2]\right),$$

and then taking logarithm of both sides yields

$$\mathbb{E}\left[\max_{i\in[K]} X_i^2\right] \leq \frac{1}{\lambda}\log K + 16\lambda \max_{i\in[K]} \sigma_i^4 + \max_{i\in[K]} \mathbb{E}[X_i^2].$$

Minimizing the RHS above over $\lambda \in \left(0, \frac{1}{4\max_{i\in[K]}\sigma_i^2}\right]$ gives optimal $\lambda^* = \frac{1}{4\max_{i\in[K]}\sigma_i^2}$, and then plugging in $\lambda^*$ yields

$$\mathbb{E}\left[\max_{i\in[K]} X_i^2\right] \leq 4\left(\log K + 1\right)\max_{i\in[K]} \sigma_i^2 + \max_{i\in[K]} \mathbb{E}[X_i^2]$$
$$\leq \left(4\log K + 5\right)\max_{i\in[K]} \sigma_i^2 + \max_{i\in[K]} \nu_i^2$$

where the last inequality uses

$$\max_{i\in[K]} \mathbb{E}[X_i^2] \leq \max_{i\in[K]}\left[\text{Var}(X_i) + \nu_i^2\right] \leq \max_{i\in[K]} \sigma_i^2 + \max_{i\in[K]} \nu_i^2.$$

This completes the proof. $\qquad\square$

**Lemma 9** (Appendix B in Honorio and Jaakkola [2014]). *Let $X$ be a sub-Gaussian random variable with mean $\nu$ and variance proxy $\sigma^2$. For $|\lambda| \leq \frac{1}{4\sigma^2}$,*

$$\mathbb{E}\left[e^{\lambda\left(X^2 - \mathbb{E}[X^2]\right)}\right] \leq e^{16\lambda^2\sigma^4}.$$

# B   Lemmas and facts from Russo and Van Roy [2016]

We use the following lemmas and facts from Russo and Van Roy [2016] in the proof of Theorem 2.

**Lemma 10** (Lemma 3 in Russo and Van Roy [2016]). *Suppose there is a $H_t$ measurable random variable $\gamma$ such that for any $a \in \mathcal{A}$, $R_{t+1,a}$ is $\gamma$ sub-Gaussian conditional on $H_t$. Then for any $t = 0, 1, \ldots$ and $a, a_* \in \mathcal{A}$, with probability one,*

$$\mathbb{E}_t[R_{t+1,a}|A_* = a_*] - \mathbb{E}_t[R_{t+1,a}] \leq \gamma\sqrt{2\mathbf{d}_{\text{KL}}\left(\mathbb{P}_t(R_{t+1,a} \in \cdot|A_* = a_*)\|\mathbb{P}_t(R_{t+1,a} \in \cdot)\right)}.$$

**Fact 2** (Fact 5 in Russo and Van Roy [2016]: chain rule of mutual information). *The mutual information between a random variable $X$ and a collection of random variables $(Z_1, \ldots, Z_T)$ can be written as,*

$$\mathbb{I}\left(X; (Z_1, \ldots, Z_T)\right) = \mathbb{I}\left(X; Z_1\right) + \mathbb{I}\left(X; Z_2|Z_1\right) + \cdots + \mathbb{I}\left(X; Z_T|Z_1, \ldots, Z_T\right).$$

**Fact 3** (Fact 6 in Russo and Van Roy [2016]: KL divergence form of mutual information). *The mutual information between random variables $X$ and $Y$ can be written as,*

$$\begin{aligned}\mathbb{I}(X;Y) &= \mathbb{E}_X\left[\mathbf{d}_{\text{KL}}\left(\mathbb{P}(Y \in \cdot|X) \,\|\, \mathbb{P}(Y \in \cdot)\right)\right] \\ &= \sum_{x\in\mathcal{X}} \mathbb{P}(X = x)\mathbf{d}_{\text{KL}}\left(\mathbb{P}(Y \in \cdot|X = x) \,\|\, \mathbb{P}(Y \in \cdot)\right).\end{aligned}$$

**Fact 4** (Fact 10 in Russo and Van Roy [2016]). *For any matrix $M \in \mathbb{R}^{d\times d}$,*

$$\text{Trace}\left(M\right) \leq \sqrt{\text{Rank}(M)}\|M\|_{\text{F}}$$

*where $\|M\|_{\text{F}}$ is the Frobenius norm of $M$.*