# OpenReview forum: "An Analysis of Ensemble Sampling"
_NeurIPS.cc/2022/Conference — NeurIPS 2022 Accept_

### Official Review · Reviewer_gXhC · 2022-07-10

**Rating:** 5
**Confidence:** 2
**Soundness:** 3 good
**Presentation:** 3 good
**Contribution:** 3 good

**Summary:**

This paper aims to establish a regret bound for ensemble sampling when it is applied to the linear bandit problem. This is done through leveraging information-theoretic concepts and novel analytic techniques.

Specifically, this paper:

* Formulates the linear bandit problem and the ensemble sampling algorithm
* Provides a bayesian regret bound for ensemble sampling in linear bandits
* Motivate the general regret bound and sketch the proof

**Questions:**

Could the authors include some more detailed instructions about how this could be applied to downstream tasks?

**Limitations:**

In the conclusion section, the authors mentioned several ways the results of this paper can be used, including analyzing and approximating different sampling algorithms in different problems. It would be great if the authors could elaborate on that a little bit more.

**Strengths And Weaknesses:**

Strengths:
This paper is the first rigorous regret analysis of ensemble learning, and it provides a bayesian regret bound to it in linear bandits.

Weaknesses:
This is likely just personal taste and limitation, but it would be great if the authors could include some instructions about how this could be applied to downstream tasks like neural architecture design.

---

> ### Author Response · Authors · 2022-08-02
> **Responses to Reviewer gXhC**
>
> We thank the reviewer for reviewing and recommending to accept this paper. Our point-to-point responses to your concerns are as follows:
>
> **"This is likely just personal taste and limitation, but it would be great if the authors could include some instructions about how this could be applied to downstream tasks like neural architecture design."**
>
> This is a theoretical paper whose main contribution is to provide the first rigorous analysis of ES. As we have mentioned in the introduction, a growing literature [Osband et al., 2016, Russo et al., 2018, Osband et al., 2018, Lu et al., 2018, Osband et al., 2019, Dwaracherla et al. 2020, Osband et al., 2022] has applied variants of ES to a broad class of online learning and reinforcement learning problems. In many of these papers, the particles in ES are chosen to be (deep) neural networks.
>
> Neural architecture search (design) is an interesting and exciting research field. Recently, there are papers trying to solve this problem based on reinforcement learning. We believe that ES can also be used in this field, especially for problems that have complex approximation architecture and require exploration.
>
> **"In the conclusion section, the authors mentioned several ways the results of this paper can
> be used, including analyzing and approximating different sampling algorithms in different
> problems. It would be great if the authors could elaborate on that a little bit more."**
>
> Please refer to "responses to all reviewers".

---

### Official Review · Reviewer_iQHa · 2022-07-11

**Rating:** 8
**Confidence:** 4
**Soundness:** 4 excellent
**Presentation:** 4 excellent
**Contribution:** 3 good

**Summary:**

The paper develops a regret bound for the methodology of ensemble sampling as applied to Linear Gaussian Bandits.  The resulting bound shows the regret to be a combination of the regret of Thomson sampling and a term to capture the posterior distribution mismatch.
In order to prove this regret, they prove a general regret bound for linear bandits for any learning algorithm which bounds the regret by a combination of the entropy of the optimal solution and a gap between the model and exact and approximate posterior as observed from sampling. They then use this result for their own bound.

**Questions:**

It would be good to see a comparison of regret bounds for other learning methods on the linear gaussian bandits besides Thomson sampling. Though part of the bound provided by them is order optimal, it is difficult to judge the quality of the overall bound without such a discussion.


**Limitations:**

The authors have not addressed this issue. I do not think the work has any direct negative societal impact.
The key limitation of the work is the use of Linear Bandits which seems to be a requirement of the analysis anyways.

**Strengths And Weaknesses:**

Strengths
The paper provides the first theoretical results for ensemble sampling.
It also provides intermediate bounds which can be leveraged by other learning methods to prove their own bounds.

Weaknesses
The paper focuses on linear bandits which are relatively simple.

---

> ### Author Response · Authors · 2022-08-02
> **Responses to Reviewer iQHa**
>
> We thank the reviewer for the thorough, thoughtful and insightful feedback. In terms of problems beyond linear bandits, see "responses to all reviewers".
>
> We compare the regret bound of ES with the regret bounds of upper confidence bound (UCB), TS,  and information-directed sampling on the linear gaussian bandits with focus on the finite action set studied in the paper. For UCB, Abbasi-Yadkori et al. [2011] and Lattimore and Szepesvari [2020] show that it achieves the worst-case regret bound $O(d\sqrt{T}\log(T))$ (for infinite action set), and we believe the worst-case regret bound for finite action set has the improvement of a factor of $\sqrt{d}$. For TS, Russo and Van Roy [2016] proves that TS achieves the Bayesian regret bound $O(\sqrt{dT\mathbb{H}(A_*)})$ where $\mathbb{H}(A_*)$ is the entropy of the optimal action $A_*$. This bound improves upon the worst-case bound of UCB since $\mathbb{H}(A_*)$ can be much smaller than $\log(K)$ and $\log(T)$ when $T\geq K$. Russo and Van Roy [2020] shows that IDS achieves the same Bayesian bound as TS. We will further clarify these comparisons in the revision.
>
> [1] Abbasi-Yadori et al. Improved algorithms for linear stochastic bandits, NeurIPS 2011.
>
> [2] Lattimore and Szepesvari, Bandit algorithms, 2020.
>
> [3] Russo and Van Roy, An information-theoretic analysis of Thompson sampling, JMLR 2016.
>
> [4] Russo and Van Roy, Learning to optimize via information-directed sampling, Operations Research 2018.

---

### Official Review · Reviewer_oHyo · 2022-07-11

**Rating:** 7
**Confidence:** 3
**Soundness:** 4 excellent
**Presentation:** 4 excellent
**Contribution:** 2 fair

**Summary:**

The paper focuses on the Ensemble Sampling (ES) algorithm, a method designed to approximate Thompson Sampling (TS) in settings where the exact posterior on unknown parameters is intractable. ES maintains an ensemble of plausible models and instead of attempting to sample a model from the full intractable posterior, it samples from a discrete distribution over the ensemble. It then proceeds, similarly to TS, to select at action which is optimal under the assumption that the sampled model represents the ground truth. The main contribution of the paper is to present a rigorous Bayesian regret analysis of the algorithm in the stochastic linear bandit setting, where Lu and van Roy (2017) previously presented a frequentist regret analysis that has subsequently been identified to be flawed.

The paper produces an ES-specific analysis by first deriving a Bayesian regret bound for ‘any learning algorithm’ which essentially consists of the information theoretic bounds of Russo and Van Roy (2016) for exact TS plus a term which is proportional to the cumulative differences between the action selection distribution of TS and of the alternative learning algorithm which is the subject of the analysis.


**Questions:**

General: Kveton et al. (2020, UAI) present a result of a similar flavour, but in the (stronger) frequentist setting. They have (as I understand it) a bound on the (frequentist) regret of any algorithm for stochastic linear bandits, which can then be specified to particular algorithms by deriving the appropriate form of a general term. I think this is an important related piece of work which should be addressed in the present paper. My question is how does the new analysis of the Bayesian regret compare to that of the frequentist regret. Does it open a possibility to analyse algorithms for which the result of Kveton et al. (2020) cannot be used? Does it just provide a quicker route to a reasonable performance guarantee for randomised algorithms? Could a stronger analysis of ES be provided using the framework from Kveton et al. (2020)? Could the present analysis inspire some analysis in some more challenging setting where the frequentist analysis could not be so readily extended?

References:
-	Kveton, Szepesvari, Ghavamadzeh, Boutilier (2020) Perturbed-History Exploration in Stochastic Linear Bandits. In Unceratinty in Artificial Intelligence


**Limitations:**

Adequately addressed, there are unlikely to be ethical issues linked to this theoretical work.

**Strengths And Weaknesses:**

The paper is well-written, in straight-forward language and clearly presents its contributions. ES is, as the related literature attests, a practical and relevant alternative to TS and thus any valid analysis of its performance is useful, particularly since previous results have been shown to be invalid. The theoretical results are, as far as I have been able to verify, accurate. As the authors suggest, there does seem to be opportunity to extend this line of analysis to more complex problems.

In my opinion the weaknesses are twofold. Firstly, as I describe in more detail below, I feel closely related results giving bounds on frequentist regret in stochastic linear bandits are not adequately addressed. Secondly, I feel the paper could have been stronger if it attempted to push beyond the linear bandit setting, where as the authors identify, ES as an approximation to TS is not necessary and the utility of the analysis is principally illustrative.

On this second point, I feel the paper could have been stronger if it had at least speculated in more detail as to what an analysis of a more challenging setting would look like or as to what the principal theoretical challenges would entail. Ideally, the paper would go so far as to extend this analytical technique to at least one setting where ES is genuinely required to deploy a TS-like approach. The level of contribution that merits publication is of course a subjective matter, and I am therefore not bound to the view that it is close to the borderline for NeurIPS. I will pay close attention to the subsequent discussions with authors and other reviewers to update my recommendation on this point.

---

> ### Author Response · Authors · 2022-08-02
> **Responses to Reviewer oHyo**
>
> We thank the reviewer for the thorough, thoughtful and insightful feedback. In terms of more complex settings beyond linear bandits, see "responses to all reviewers".
>
> We thank the reviewer for bringing up Kveton et al. (2020). Our general regret bound is very different from that in Section 4.2 of Kveton et al. (2020). The general regret bound in Kveton et al. (2020) is tailored to analyzing randomized algorithms with good concentration and anti-concentration properties (see Equations (7,8) and the one 3 lines above Eqn. (7)). Since ES uses $M$ models to approximate the posterior, due to this discrete nature of ES, it is not sure whether ES enjoys such concentration and anti-concentration properties. In other words, the probability constants $p_1, p_2, p_3$ in Section 4.2 of Kveton et al. (2020) could be really bad for ES. In addition, ES is an incremental algorithm instead of adding noises at each timestep, which makes the proof template in Kveton et al. (2020) does not directly fit. Furthermore, Kveton et al. (2020) studies the problems with bounded rewards, while we study the unbounded case. We will further clarify these issues in the revision.

---

> > ### Comment · Reviewer_oHyo · 2022-08-04
> > **Thank you**
> >
> > Hi authors,
> >
> > Thank you for this explanation with regards to the differences between your analysis and that of Kveton et al. (2020). I wonder whether you plan to add some of these remarks to the final version of your paper? I think it would be beneficial to do so.

---

> > > ### Author Response · Authors · 2022-08-04
> > > **Thank you again!**
> > >
> > > Thank you again for bringing up this relevant paper. We will definitely cite it and add some remarks to illustrate the difference in the final version of our paper. In particular, we will clarify that Kveton et al. (2020) has derived a general regret bound for algorithms with good concentration and anti-concentration properties, but it could not be directly applied to analyzing ES due to ES's discrete and incremental update nature.

---

> > > > ### Comment · Reviewer_oHyo · 2022-08-08
> > > > **Increase Score**
> > > >
> > > > Thanks for confirming, in light of that and the general response to all reviewers I will increase my score. Congrats on the strong paper.

---

> > > > > ### Author Response · Authors · 2022-08-09
> > > > > **Thank you very much!**
> > > > >
> > > > > Thank you very much for increasing the score and for the thorough, thoughtful and insightful feedback!

---

### Official Review · Reviewer_MPVW · 2022-07-12

**Rating:** 5
**Confidence:** 3
**Soundness:** 3 good
**Presentation:** 3 good
**Contribution:** 3 good

**Summary:**

The authors propose a novel analysis of Thompson Sampling using ensemble sampling, which is an approximation technique in order to correct for the fact that in general, sampling from the posterior distribution of the unknown vector theta is impractical in high dimensions. The idea of ensemble sampling resembles a particle filter, where one samples a finite number of points in order to approximate the intractable posterior distribution. They propose a regret analysis which corrects deficiencies of an earlier analysis of (Lu and Van Roy, 2017).

**Questions:**

Two questions:
- is it possible to lower bound the value of $M$ as a function of $K,T$ to ensure a regret comparable to Thompson Sampling ?
- why is there a noise floor in the regret upper bounds ? and can it be fixed ?

**Limitations:**

See above.

**Strengths And Weaknesses:**

Strengths:
- The paper is well written and easy to follow.
- The paper corrects technical deficiencies in an earlier paper
- Theorem 2 is nice in the sense that it might apply to a wider range of settings, and shows that controlling the Hellinger distance between the approximate distribution and the actual posterior yields usable upper bounds.
- In general, the idea of finding computationally efficient alternatives to Thompson Sampling for structured bandits is a very important problem, and certainly non trivial.

Weaknesses
- Dependency on the number of arms $K$: The dependency on the regret on the number of points $M$ is very significant, and imposes that, for the regret bound to be non trivial, one requires at least $M \ge K$, and for the regret bound to be close to the regret of Thompson Sampling one requires $M \ge {T K \over d}$. Now, the number of arms $K$ is going to be larger than the dimension $d$ (otherwise one can simply ignore the linear structure and treat the problem like a classical $K$ armed bandit, and in many cases, $K$ is going to be exponential in $d$ (an example of this would be combinatorial bandits). This implies that the algorithm might not be usable in high dimensions. The authors argue that one could use $M = O( {d \over \epsilon^2})$ instead of  $M = O({K \over \epsilon^2})$ but this seems dubious. At the very least the authors could expand on this issue. In fact, determining a lower bound on $M$ as a function of $K,T$ would be very interesting.

- Noise floor when $\sigma^2 \to 0$: in the regret upper bound of Theorem 1, when the variance of the rewards $\sigma^2$ is taken close to $0$, the regret upper bound becomes $O( \sqrt{dT H(A)})$, which is probably not tight since, when rewards have no variance, it is sufficient to sample $d$ linearly independent actions in order to figure out the optimal action with probability 1, and hence the regret in this regime should be at most $O( d \sqrt{\log K})$ where $O(\sqrt{\log K})$ is the expected value of the largest gap between the optimal arm and a non optimal arm.

---

> ### Author Response · Authors · 2022-08-02
> **Responses to Reviewer MPVW**
>
> We thank the reviewer for the thorough, thoughtful and insightful feedback.
>
> To clarify, the algorithm can be used in the high-dimensional problems; however, the second term in the current regret bound might not be tight. We conjecture that it can be improved from $O(K/\epsilon^2)$ to $O(d/\epsilon^2)$. The reason why it is not tight is that we use Lemma 11 in Lu and Van Roy [2017], which relies on the method of (finite) types and results in the dependence on $K$. It requires new techniques to further improve this bound and we leave it to future work. That said, we believe that the direction of future research is to further improve the upper bound of $M$, i.e., the sufficient condition of $M$. In terms of the lower bound of $M$, we conjecture it might be a function of $d$ and $T$. Though currently we don not know how to improve Theorem 1, we believe Theorem 2, which has independent interest, might be further improved. If we change the learning target from optimal action $A_*$ to the coefficient vector $\theta$, it is possible to prove a generalization of Theorem 2 that applies to infinite action set.
>
> In terms of noise floor when $\sigma^2\to 0$, we fully agree that when $\sigma = 0$ (the deterministic case), the optimal regret should be independent of $T$.  Our regret bound is not tight in this case. This is not surprising since most existing analyses developed for stochastic bandits do not achieve tight regret bounds in deterministic bandits. On the other hand, when $\sigma>0$, even if $\sigma$ is very small, the regret is expected to scale as $\sqrt{T}$ instead of a constant (see Dani et al. [2017] and Lattimore and Szepesvari [2020]), due to the cost of exploration in stochastic bandits.
>
> [1] Dani et al., The price of bandit information for online optimization. NeurIPS 2007.
>
> [2] Lattimore and Szepesvari, Bandit algorithms, 2020.

---

### Author Response · Authors · 2022-08-02
**Responses to all reviewers**

We would like to thank all reviewers for reviewing our paper. We very much appreciate it that all reviewers recommend to accept this paper. Several reviewers have raised questions about potential extension of the current analysis beyond linear bandits, which we will address below. We will also provide point-to-point response to other questions raised by reviewers in separate responses to them.

**Significance of this paper and potential extensions beyond linear bandits**

We would like to clarify that this paper is the first one that provides a rigorous analysis of ensemble sampling (ES), a widely used approximate Thompson sampling algorithm. Even though the analysis in this paper is restricted to linear bandits, we believe that it is significant since it not only has fixed a long-standing bug but also offers novel insights on how to analyze ES and even general approximate TS algorithms for a broader class of bandit problems, as detailed below.

We believe that Theorem 2 in this paper could be generalized to any bandit problem and any learning algorithm for it based on the concept of information ratio proposed in Russo and Van Roy [2016], as long as the information ratio can be appropriately bounded. These problems include online learning with full information feedback and graph information feedback, sparse linear bandits, combinatorial semi-bandits and pure exploration problems. Variants of Theorem 2 are particularly useful for analyzing approximate TS since the regret bounds are based on the Hellinger distance between the algorithm's sampling distribution and the posterior distribution of the optimal action. We conjecture that hypermodels and Langevin TS could be analyzed based on Theorem 2. In addition, Theorem 1 for ES in this paper can be generalized to bandit problems with linear Gaussian structure and more complex action space, such as combinatorial semi-bandits with linear generalization (Wen et al. [2015]) and linear cascading bandits (Zong et al. [2016]).

[1] Wen et al., Efficient learning in large-scale combinatorial semi-bandits, ICML 2015.

[2] Zong et al., Cascading bandits for large-scale recommendation problems, UAI 2016.

---

### Meta-Review · Area_Chair_JLQg · 2022-08-25

**Recommendation:** Accept
**Confidence:** Certain

**Metareview:**

There was broad agreement about this paper. All reviewers valued the main contribution: the first rigorous analysis of ensemble sampling as an approximation to Thompson sampling. The reviews and discussion all indicate that the paper is clear and honest in stating its contributions and supporting them with evidence, and that this main contribution is valuable. The remaining discussion revolved around finer points of scope of significance; on one hand, the setting of linear bandits was viewed as relatively simple, and some of the given bounds may not be tight; on the other hand, there is speculation that some of the analysis may apply more broadly than the simple setting considered in the paper. Overall, there is broad consensus about recommending the paper be accepted.

**Award:**

No

---

### Decision · Program_Chairs · 2022-09-14

Accept